# Bias in comparisons of mortality among very preterm births: A cohort study

**Amélie Boutin**[1¤]*, **Sarka Lisonkova**[1], **Giulia M. Muraca**[1,2], **Neda Razaz**[2], **Shiliang Liu**[3], **Michael S. Kramer**[4], **K. S. Joseph**[1,5]

**1** Department of Obstetrics and Gynaecology, BC Children's and Women's Hospital and Health Centre, and the University of British Columbia, Vancouver, British Columbia, Canada, **2** Clinical Epidemiology Unit, Karolinska Institutet, Stockholm, Sweden, **3** Maternal and Infant Health Section, Public Health Agency of Canada, Ottawa, Ontario, Canada, **4** Departments of Pediatrics and of Epidemiology and Biostatistics, McGill University, Montreal, Quebec, Canada, **5** School of Population and Public Health, University of British Columbia, Vancouver, British Columbia, Canada

¤ Current address: Department of Pediatrics, Université Laval, Reproduction, Mother and Youth Health & Population Health and Optimal Health Practices Units, CHU de Québec-Université Laval Research Center, Quebec City, Canada

* aboutin@bcchr.ca

**Data Availability Statement:** U.S. data are publicly available from the National Center for Health Statistics (https://www.cdc.gov/nchs/data_access/vitalstatsonline.htm). Canadian data cannot be shared publicly because of Non-Disclosure/

## Abstract

### Background

Several studies of prenatal determinants and neonatal morbidity and mortality among very preterm births have resulted in unexpected and paradoxical findings. We aimed to compare perinatal death rates among cohorts of very preterm births (24–31 weeks) with rates among all births in these groups (≥24 weeks), using births-based and fetuses-at-risk formulations.

### Methods

We conducted a cohort study of singleton live births and stillbirths ≥24 weeks' gestation using population-based data from the United States and Canada (2006–2015). We contrasted rates of perinatal death between women with or without hypertensive disorders, between maternal races, and between births in Canada vs the United States.

### Results

Births-based perinatal death rates at 24–31 weeks were lower among hypertensive than among non-hypertensive women (rate ratio [RR] 0.67, 95% CI 0.65–0.68), among Black mothers compared with White mothers (RR 0.94, 95%CI 0.92–0.95) and among births in the United States compared with Canada (RR 0.74, 95%CI 0.71–0.75).

However, overall (≥24 weeks) perinatal death rates were higher among births to hypertensive vs non-hypertensive women (RR 2.14, 95%CI 2.10–2.17), Black vs White mothers (RR 1.86, 95%CI 184–1.88;) and births in the United States vs Canada (RR 1.08, 95%CI 1.05–1.10), as were perinatal death rates based on fetuses-at-risk at 24–31 weeks (RR for hypertensive disorders: 2.58, 95%CI 2.53–2.63; RR for Black vs White ethnicity: 2.29, 95% CI 2.25–2.32; RR for United States vs Canada: 1.27, 95%CI 1.22–1.30).

Confidentiality Agreement. Data are available from the Canadian Institute for Health Information (https://www.cihi.ca/en/access-data-and-reports/make-a-data-request) for researchers who meet the criteria for access to confidential data.

**Funding:** This work was supported by grants from the Canadian Institutes of Health Research (PER-150902 and F17-02161) to KSJ and SL, a Killam Postdoctoral Research Fellowship Award and a Fellowship Award from the Canadian Institutes of Health Research to AB, a Michael Smith Scholar Award to SL, a Fellowship Award from the Canadian Institutes of Health Research to GM, an Investigator Award from the British Columbia Children's Hospital Research Institute to KSJ. The funders had no role in study design, data collection and analysis, decision to publish, or preparation of the manuscript.

**Competing interests:** The authors have declared that no competing interests exist.

## Conclusion

Studies of prenatal risk factors and between-centre or between-country comparisons of perinatal mortality bias causal inferences when restricted to truncated cohorts of very preterm births.

## Introduction

Preterm birth is a major health problem worldwide, associated with high mortality and morbidity and life-long disability [1–3]. As a consequence, several research initiatives have targeted preterm birth. The Canadian Neonatal Network and other groups have conducted many studies among preterm populations, focusing on prenatal risk factors for perinatal outcomes, or comparing perinatal and neonatal outcomes by health centre and country, in addition to studies of interventions following preterm birth [4–13].

Studies of risk factors for adverse perinatal outcomes are typically designed to identify the potentially modifiable underlying causes of these outcomes. Similarly, comparisons of neonatal morbidity and mortality by centre or country can highlight regional and other disparities and stimulate initiatives to improve maternal and neonatal health care. Several reports document reductions in adverse neonatal outcomes at the national level following programs to foster optimal clinical practices identified through between-centre comparisons of neonatal morbidity and mortality among very preterm births [14, 15].

Despite the apparent utility of the above-mentioned investigations, many studies restricted to preterm infants have shown unexpected results. For instance, studies have reported higher neonatal mortality among very preterm infants born to normotensive mothers compared with those born to hypertensive mothers [6], and better survival among very preterm infants of older mothers than among those born to younger mothers [5]. These findings highlight the paradox of intersecting perinatal mortality curves, a phenomenon described by Yerushalmy over 50 years ago [16]: low birth weight infants of mothers who smoke in pregnancy have higher neonatal survival than low birth weight infants of non-smoking mothers, whereas the opposite is observed at higher birth weights. This paradox is a general phenomenon that occurs across diverse contrasts by risk factor (e.g., multifetal pregnancies, maternal age) and outcome (e.g., stillbirth, neonatal death, cerebral palsy, and sudden infant death syndrome), irrespective of how "maturity" is defined (by birth weight or gestational age). Despite a lack of agreement on the mechanism responsible for the paradox [16–27], the risk factors across which the paradox has been observed (e.g., maternal smoking and hypertension) are recognized as deleterious to fetal and infant health.

Although the paradox of intersecting curves has received extensive attention in the epidemiologic literature, its wider implications for etiologic studies and geographic comparisons restricted to very preterm births have often been overlooked. In this study, we analyze the effects of prenatal exposures on perinatal outcomes. To illustrate the consequences and highlight the risk of bias of restricting studies to very preterm births, we examined the association between 1) a pregnancy risk factor (i.e., hypertensive disorders in pregnancy), 2) a social determinant of health (i.e., race), or 3) regions (i.e., Canada vs. United States), and perinatal mortality.

## Methods

Our cohort study was based on all live births and stillbirths in the United States and Canada for the years 2006–2015. Data on births in the United States were obtained from the period

linked birth/infant death files and fetal death files of the National Center for Health Statistics [28], which together include information from live birth and stillbirth registrations in the United States. Data for births in Canada were obtained from the Discharge Abstract Database of the Canadian Institute for Health Information [29], which includes all delivery hospitalizations and represents approximately 98% of births in Canada (excluding Quebec). Gestational age in both data sources was based on the clinical estimate of gestation. Information in these databases has been validated and is routinely used in epidemiologic studies [30, 31].

The study population was restricted to singleton births with a clinical estimate of gestation ≥24 weeks, as births at earlier gestational age are associated with wide variations in resuscitation, active treatment and birth registration practices [32–34]. Fetuses or infants with congenital or chromosomal anomalies (including anencephaly, meningomyelocele/spina bifida, cyanotic congenital heart disease, congenital diaphragmatic hernia, omphalocele, gastroschisis, limb reduction defect, cleft lip or palate, Down syndrome, hypospadias or suspected chromosomal disorder) were excluded.

### Independent variables of interest

Women with hypertensive disorders in pregnancy included those with chronic (pre-pregnancy) hypertension, gestational hypertension and eclampsia, while non-hypertensive women included women without any of the aforementioned conditions. Maternal race comprised four categories: White, Black, Native American, and Asian (including Pacific Islanders). Whites were the reference group in all comparative analyses. Analyses of hypertensive disorders in pregnancy and maternal ethnicity were restricted to births in the United States. The regional comparisons contrasted births at ≥24 weeks in Canada vs. the United States.

### Dependant variable

The primary outcome was perinatal death, defined as stillbirth or early neonatal death (i.e., death within 7 days of delivery). The secondary outcome was early neonatal death.

### Methods for the calculation of rates

Overall and gestational age-specific rates of perinatal death were calculated using two different denominators: births-based and fetuses-at-risk [35]. Gestational age-specific rates under the births-based formulation were calculated as the proportion of perinatal deaths among total births (live births plus stillbirths) at a specific gestational week. Under the fetuses-at-risk formulation, gestational age-specific perinatal death rates were calculated using the number of perinatal deaths at each completed gestational week in the numerator, and the number of fetuses at risk for perinatal death at the beginning of the gestational week in the denominator (i.e., the number of fetuses who delivered at or after the gestational week in question). For instance, the gestational age-specific perinatal death rate at 28 weeks' gestation was calculated using the number of perinatal deaths at 28 weeks in the numerator and the number of fetuses in utero at the beginning of the 28th week of gestation as the denominator (including those born at 28 weeks and all those delivered at a later gestational age) [35]. The gestational age-specific perinatal death rates under the fetuses-at-risk formulation can be described as a cumulative risk over a one-week period and approximate the conditional hazard rate.

Overall (≥24 weeks) births-based and fetuses-at-risk rates of perinatal death are equivalent, since both numerators include all deaths at ≥24 weeks and both denominators include all births at ≥24 weeks. Birth-based and fetuses-at-risk rates of early neonatal death were also calculated in a similar manner. Gestational age-specific rates under the birth-based approach were calculated as the proportion of early neonatal deaths among live births at a specific

gestational week, while fetuses-at-risk calculations involved dividing the number of early neonatal deaths at a given gestational week by the number of fetuses at risk of birth and early neonatal death at that gestational week.

We carried out additional analyses to assess the potential role of confounding in contrasts between singletons of women with or without hypertensive disorders in pregnancy as the relation with perinatal death could be confounded by maternal age, race, comorbidity or other factors. Since the variables available for adjustment were limited, this supplementary analysis was intended to gauge the degree of potential confounding by putative confounders, and to assess whether such confounding would alter effect estimates modestly or reverse the direction of the association. Logistic regression was used to contrast hypertensive vs. non-hypertensive women with regard to perinatal death after adjusting for maternal age (indicator variables for 5-year strata), race, and diabetes.

In a final analysis, designed to illustrate differences between the denominators used in the births-based and fetuses-at-risk formulations of gestational age-specific perinatal mortality, we contrasted the compared categories (women with vs. without hypertensive disorders, White women vs. women of other race and women from Canada vs. the United States) in terms of fetuses-at-risk birth rates. These birth rates were calculated using births at any gestational week in the numerator and fetuses at risk of birth in the denominator.

The study received ethics approval from the institutional review board at the University of British Columbia. All analyses were carried out using SAS statistical software (version 9.4, SAS Institute Inc., Cary, NC, USA).

## Results

The United States data sources included 39,298,721 eligible singleton live births and stillbirths ≥24 weeks' gestation between 2006 and 2015 (Table 1). We identified 2,715,169 hospital deliveries of eligible singletons between 2006 and 2015 in Canada (excluding Quebec).

**Table 1. Numbers of singleton births and perinatal deaths (excluding congenital and chromosomal anomalies), in the United States, 2006–2015.**

|  | Overall | Hypertensive disorders in pregnancy | | Maternal race | | | |
|---|---|---|---|---|---|---|---|
|  |  | No | Yes | White | Black | Native American | Asian |
| **Women ≥35 years old at delivery** | 5,698,555 (14.5) | 5,255,189 (14.3) | 423,858 (18.7) | 4,385,076 (14.6) | 684,922 (11.0) | 40,046 (8.8) | 588,511 (23.4) |
| **Chronic hypertension** | 524,870 (1.3) | 0 (0.0) | 524,870 (23.2) | 334,853 (1.1) | 162,594 (2.6) | 7,424 (1.6) | 19,999 (0.8) |
| **Diabetes** | 2,051,552 (5.2) | 1,733,255 (4.7) | 318,297 (14.1) | 1,500,801 (5.0) | 293,453 (4.7) | 34,446 (7.6) | 222,852 (8.9) |
| **Total births ≥24 weeks** | 39,298,721 | 36,890,944 | 2,265,317 | 30,102,675 | 6,223,981 | 456,164 | 2,515,901 |
| **Total births 24–31 weeks** | 480,471 (1.2) | 382,470 (1.0) | 91,024 (4.0) | 299,506 (1.0) | 151,347 (2.4) | 5,765 (1.3) | 23,853 (0.9) |
| **Live births ≥24 weeks** | 39,161,645 | 36,777,532 | 2,249,673 | 30,011,079 | 6,187,092 | 454,391 | 2,509,083 |
| **Live births 24–31 weeks** | 422,756 (1.1) | 336,003 (0.9) | 83,511 (3.7) | 263,039 (0.9) | 133,594 (2.2) | 5,098 (1.1) | 21,025 (0.8) |
| **Stillbirth ≥24 weeks** | 137,076 | 113,412 | 15,644 | 91,596 | 36,889 | 1,773 | 6,818 |
| **Stillbirths 24–31 weeks** | 57,715 (42.1) | 46,467 (41.0) | 7,513 (48.0) | 36,467 (39.8) | 17,753 (48.1) | 667 (37.6) | 2,828 (41.5) |
| **Early neonatal deaths ≥24 weeks** | 44,555 | 39,635 | 4,424 | 31,373 | 10,383 | 593 | 2,206 |
| **Early neonatal deaths 24–31 weeks** | 21,855 (49.1) | 18,770 (47.4) | 2,813 (63.6) | 14,323 (45.7) | 6,257 (60.3) | 267 (45.0) | 1,008 (45.7) |
| **Perinatal deaths ≥24 weeks** | 181,631 | 153,047 | 20,068 | 122,969 | 47,272 | 2,366 | 9,024 |
| **Perinatal deaths 24–31 weeks** | 79,570 (43.8) | 65,237 (42.6) | 10,326 (51.5) | 50,790 (41.3) | 24,010 (50.8) | 934 (39.5) | 3,836 (42.5) |

Numbers in parentheses represent proportions (%); e.g., overall 14.5% of women were ≥35 years old at delivery, 1.3% had chronic hypertension, 5.2% had diabetes, and 1.2% of total births were 24–31 weeks' gestation.

## Hypertensive disorders in pregnancy

Table 1 presents the numbers of singleton live births, stillbirths and perinatal deaths among women with and without hypertensive disorders in the United States. Among births at 24–31 weeks' gestation, the births-based perinatal death rate was *lower* among women with hypertensive disorders (113.4 per 1,000 total/live births) compared with women without hypertensive disorders (170.6 per 1,000 total/live births; P<0.001; Table 2). On the other hand, the overall (≥24 weeks) births-based perinatal death rate among women with hypertensive disorders (8.86 per 1,000 total/live births) was significantly *higher* than among women without hypertensive disorders (4.15 per 1,000 total/live births; P<0.001; Table 2).

Fig 1 shows gestational age-specific perinatal death rates among women with vs. without hypertensive disorders calculated using births-based denominators. This contrast illustrates the paradox of intersecting perinatal mortality curves: death rates among women with hypertensive disorders were lower at early gestation but higher at later gestation than among women without hypertensive disorders.

The perinatal death rate at 24–31 weeks' gestation among women with hypertensive disorders (4.56 per 1,000 fetuses-at-risk, respectively) was significantly *higher* than the same rate

**Table 2. Comparisons of rates of perinatal death at 24–31 weeks' gestation and overall among singletons with no congenital or chromosomal anomaly, 2006–2015.**

| | Perinatal death rate at 24–31 weeks (95% CI), Births-based calculation (per 1,000 total births)[a] | RR (95% CI) | Perinatal death rate at 24–31 weeks (95% CI), Fetuses-at-risk calculation (per 1,000 fetuses-at-risk)[b] | RR (95% CI) | Perinatal death rate overall (95% CI), (per 1,000 total births)[c] | RR (95% CI) |
|---|---|---|---|---|---|---|
| **Hypertensive disorders in pregnancy**[d] | | | | | | |
| **No** | 170.6 (169.4 to 171.8) | Ref | 1.77 (1.75 to 1.78) | Ref | 4.15 (4.13 to 4.17) | Ref |
| **Yes** | 113.4 (111.4 to 115.5) | 0.67 (0.65 to 0.68) | 4.56 (4.47 to 4.65) | 2.58 (2.53 to 2.63) | 8.86 (8.74 to 8.98) | 2.14 (2.10 to 2.17) |
| | | Adj: 0.63 (0.62 to 0.64) | | Adj: 2.41 (2.36 to 2.46) | | Adj 1.92 (1.89 to 1.95) |
| **Maternal race/ ethnicity**[d] | | | | | | |
| **White** | 169.6 (168.2 to 170.9) | Ref | 1.69 (1.67 to 1.70) | Ref | 4.08 (4.06 to 4.11) | Ref |
| **Black** | 158.6 (156.8 to 160.5) | 0.94 (0.92 to 0.95) | 3.86 (3.81 to 3.91) | 2.29 (2.25 to 2.32) | 7.60 (7.53 to 7.66) | 1.86 (1.84 to 1.88) |
| **Native American** | 162.0 (152.5 to 171.5) | 0.96 (0.90 to 1.01) | 2.05 (1.92 to 2.18) | 1.21 (1.14 to 1.30) | 5.19 (4.98 to 5.40) | 1.27 (1.22 to 1.32) |
| **Asian** | 160.8 (156.2 to 165.5) | 0.95 (0.92 to 0.98) | 1.52 (1.48 to 1.57) | 0.90 (0.88 to 0.93) | 3.59 (3.51 to 3.66) | 0.88 (0.86 to 0. 09) |
| **Country of birth** | | | | | | |
| **United States** | 165.6 (164.6 to 166.7) | Ref | 2.02 (2.01 to 2.04) | Ref | 4.62 (4.60 to 4.64) | Ref |
| **Canada** | 226.0 (220.1 to 231.9) | 1.36 (1.33 to 1.40) | 1.61 (1.56 to 1.65) | 0.79 (0.77 to 0.82) | 4.29 (4.22 to 4.37) | 0.93 (0.91 to 0.95) |

[a] Births-based death rates represent proportions, with the number of perinatal deaths at 24–31 weeks in the numerator and the number of total births at 24–31 weeks in the denominator.

[b] Fetuses-at-risk rates represent cumulative incidence rates with the number of perinatal deaths at 24–31 weeks in the numerator and the number of fetuses at risk of perinatal death at 24 weeks (i.e., fetuses who were delivered at 24 weeks or later) in the denominator.

[c] This calculation is identical for births-based and fetuses-at risk formulations.

[d] Based on births in the United States.

*Adj* Adjusted for maternal age (using indicator variables for 15–19, 20–24, 25–29, 30–34, 35–39, 40–44, 45–49, 50–54 year age categories), maternal race (using indicator variables for White, Black, American Indian/Alaskan Native, Asian/Pacific Islander), and diabetes; *CI* denotes confidence intervals; *RR* denotes rate ratios.

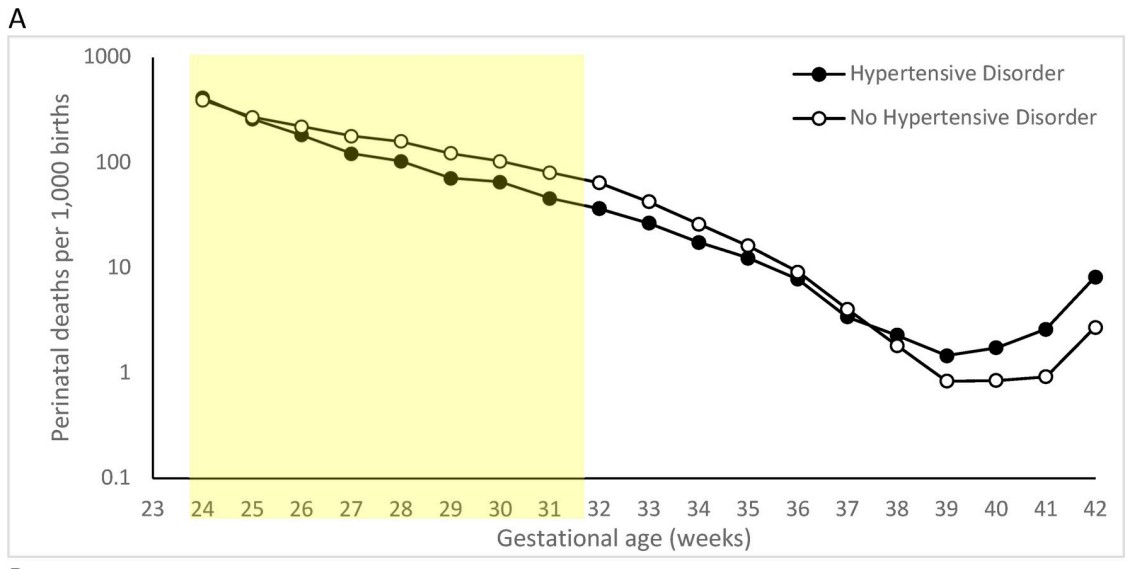

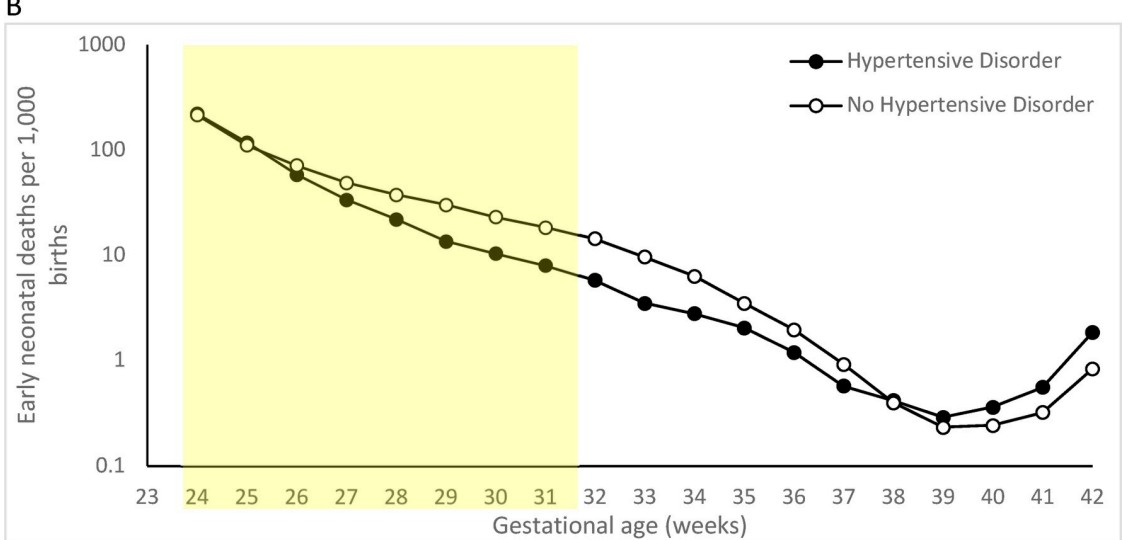

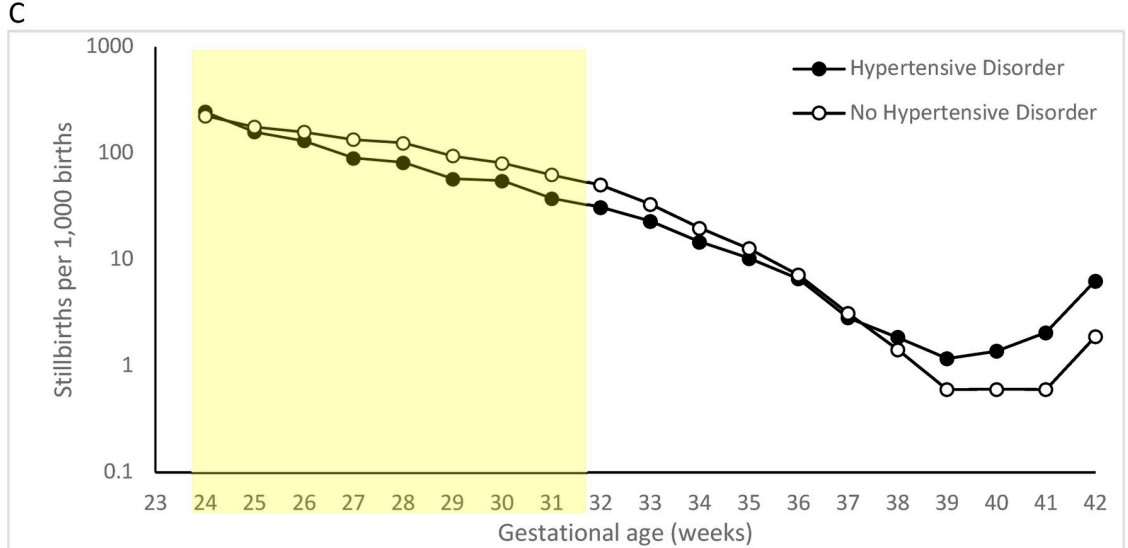

**Fig 1.** Gestational age-specific perinatal death (A), early neonatal death (B) and stillbirth (C) rates of singletons with no congenital or chromosomal anomalies among women with and without hypertensive disorders using a births-based denominator, United States, 2006–2015. The yellow area highlights the restricted subpopulation at 24–31 weeks' gestation.

among women without hypertensive disorders (1.77 per 1,000 fetuses-at-risk; P<0.001) when calculated using fetuses-at-risk denominators (Table 1). Fig 2 shows gestational age-specific perinatal mortality rates calculated based on fetuses-at-risk denominators: death rates were higher among births to women with hypertensive disorders at all gestational ages.

Adjusted odds ratios expressing the association between maternal hypertension and perinatal death differed only modestly from the unadjusted estimates (Table 2).

## Maternal racial groups

Comparisons of perinatal mortality rates among singleton births by maternal race showed the same paradox of intersecting perinatal mortality curves when calculated using births-based denominators. Birth-based perinatal death rates were significantly *lower* among singletons of Black mothers and non-significantly lower among births to Native American mothers compared with White mothers (158.6, 162.0, and 169.6 per 1,000 births, respectively; P<0.001; S1 Fig). The fetuses-at-risk formulation showed significantly *higher* perinatal death rates among Black and Native American mothers compared with White mothers (3.86, 2.05, and 1.69 per 1,000 fetuses-at-risk, respectively; P<0.001; Table 2; S1 Fig). Overall rates also showed *higher* perinatal death rates among singletons of Black and Native American mothers (Table 2). Asian mothers had significantly lower perinatal death rates than White mothers, but the difference was underestimated by births-based calculations. Comparisons of early neonatal death rates are provided in S1 Table.

## Canada vs. the United States

Among singleton births at 24–31 weeks' gestation, the births-based perinatal death rate was significantly *higher* in Canada than in the United States (226.0 vs. 165.6 per 1,000 total births; P<0.001; Table 2, S2 Fig). However, the perinatal death rate at 24–31 weeks' gestation in Canada was substantially *lower* than that in the United States when calculated using fetuses-at-risk denominators (1.61 vs. 2.02 per 1,000 fetuses-at-risk; P<0.001; Table 2, S2 Fig). Differences in the overall perinatal death rate (≥24 weeks) were similar to differences in fetuses-at-risk perinatal death rates at 24–31 weeks' gestation: overall perinatal mortality rates were significantly *lower* in Canada than in the United States (4.29 vs. 4.62 per 1,000 total births; P<0.001; Table 2).

Early neonatal death rates at 24–31 weeks in Canada and the United States similarly displayed opposite associations when calculated using births-based and fetuses-at-risk formulations (S1 Table).

## Birth rates

The birth rate at 24–31 weeks' gestation was higher among women with hypertensive disorders compared with women without hypertensive disorders (S3 Fig), among Black women compared with White women (S4 Fig) and in the United States compared with Canada (S5 Fig).

## Discussion

Our study shows that, in research with a cause-and-effect focus, comparisons of perinatal mortality between fetuses of mothers with and without hypertensive disorders of pregnancy,

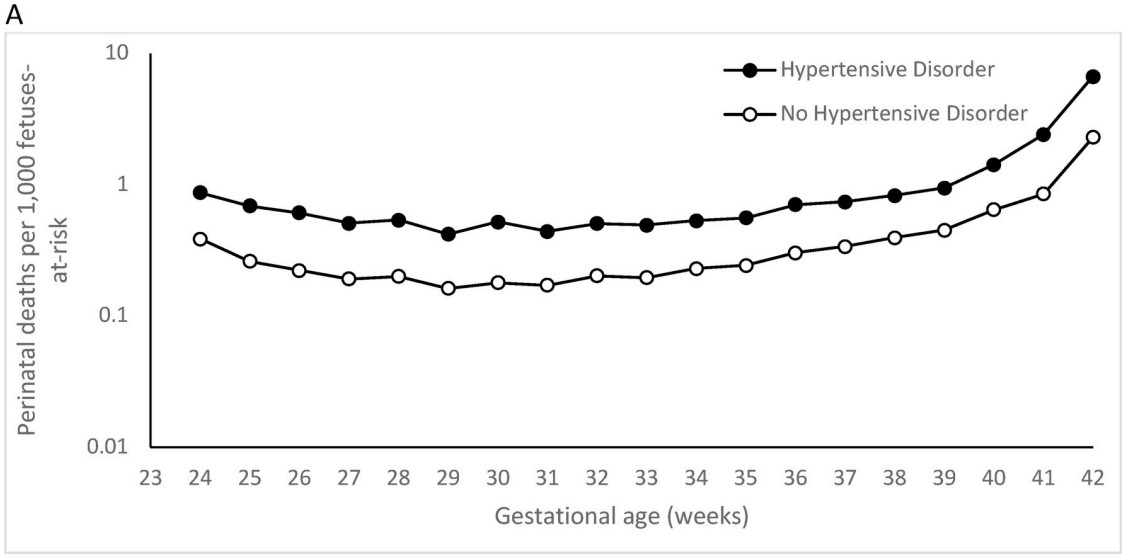

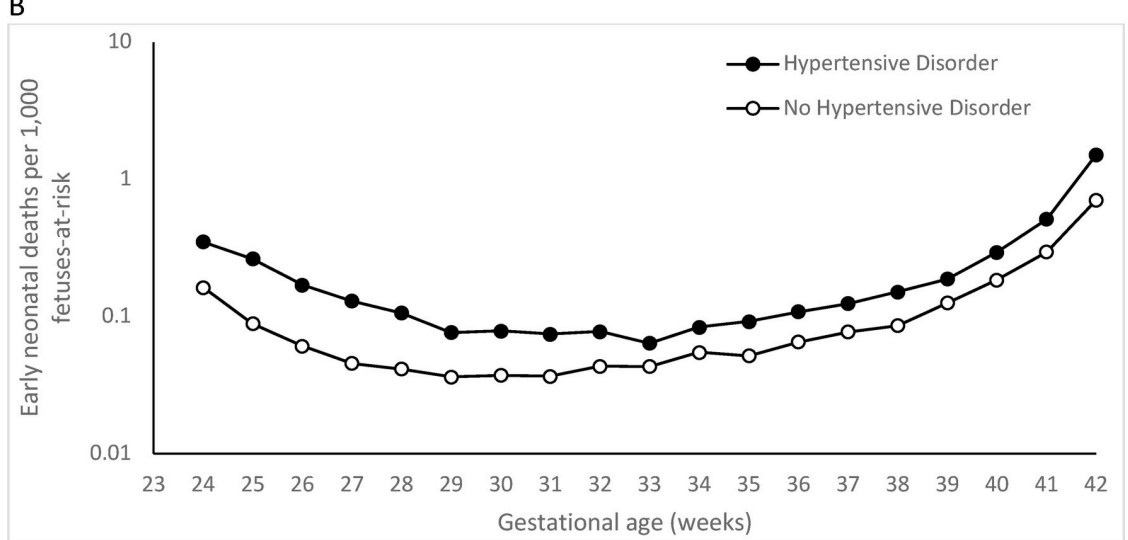

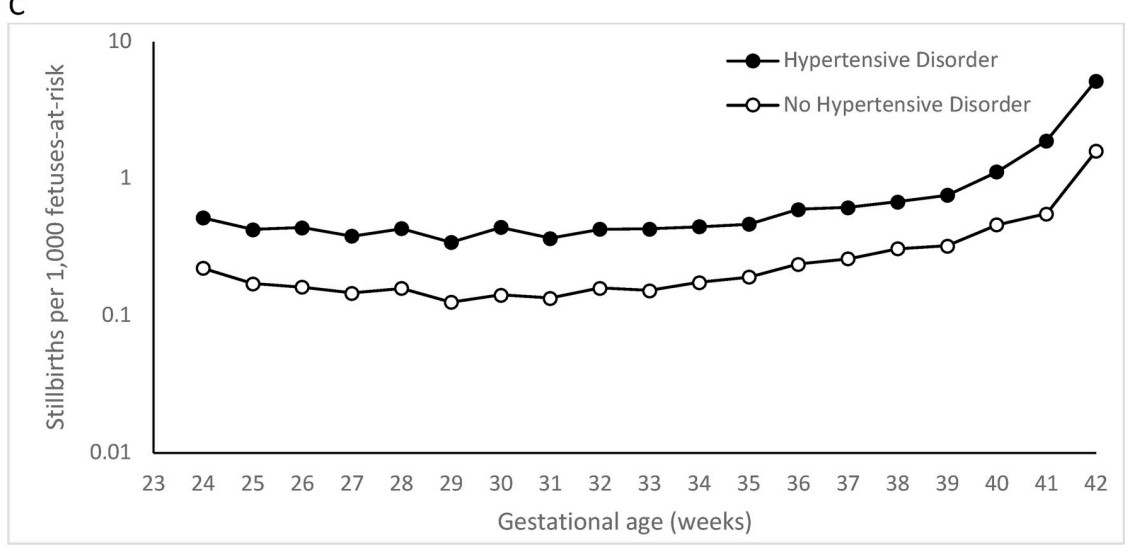

**Fig 2.** Gestational age-specific perinatal death (A), early neonatal death (B) and stillbirth (C) rates of singletons with no congenital or chromosomal anomalies among women with or without hypertensive disorders using a fetuses-at-risk denominator, United States, 2006–2015.

between mothers of different races, and between mothers in Canada and the United States, are seriously biased when the study population is restricted to very preterm births and analyses are based on births-based denominators. Births-based perinatal death rates were significantly lower among women with hypertensive disorders, among Black mothers, and among births in Canada when the study populations were restricted to very preterm births. Inferences made from such truncated births-based analyses of very preterm births conflicted starkly with those obtained from analyses of the whole cohort of fetuses (births at ≥24 weeks) and analyses based on fetuses-at-risk calculations at 24–31 week's gestation. The latter analyses showed significantly higher mortality among births to women with hypertensive disorders compared with births to non-hypertensive women, among Black and Native American mothers, and among births in the United States.

Our gestational-age specific analysis of perinatal mortality by exposure to hypertensive disorders in pregnancy, maternal race and country confirm the paradox of intersecting perinatal mortality curves. The fetuses-at-risk approach resolves the paradox by using a fetal perspective and a survival analysis formulation [35, 36]. A recent mechanistic explanation that reconciles the births-based and fetuses-at-risk approaches suggests that the lower births-based mortality experienced by higher-risk populations at earlier gestation is the consequence of an accelerated birth rate, which leads to more non-compromised fetuses being born at early gestation [37]. The problem of the paradox of perinatal mortality curves has also been framed as a collider stratification bias [19] (i.e., restricting by birth weight or gestational age, which are common effects of the perinatal determinants and unmeasured confounders, introduces a bias), and as effect-modification due to cortisol-mediated intrauterine adaptation resulting from the chronic stress associated with preterm birth [38]. However, the paradox of intersecting perinatal mortality curves manifests across a range of contrasts (including maternal age, smoking, hypertension, race, and country), and although it is possible to postulate a mechanism for the reduced mortality at preterm gestation associated with one factor or other, the need to propose a mechanism for each of these diverse contrasts supports the more parsimonious, singular explanation that the birth-based analysis is flawed.

Whereas the calculation of stillbirth rates has changed from a births-based formulation to a fetuses-at-risk denominator in recent decades (and thereby eliminated the paradox with regard to stillbirths), the situation with regard to the appropriate denominator for calculating neonatal death rates remains a topic of debate [39–44]. Some epidemiologists argue that live births (but not fetuses) experience neonatal death, and hence neonatal mortality rate calculations must use live births (and not fetuses) as the denominator [39, 41]. However, the alternative viewpoint, whereby both stillbirths and neonatal deaths are deemed to be closely related outcomes amenable to obstetric intervention, has been a traditional construct of modern obstetrics. Thus, clinical trials with a perinatal intervention (whether prenatal iron supplementation, antenatal corticosteroid therapy, magnesium sulphate for preterm birth at <31 weeks or labour induction at post-term gestation) involve the randomization of pregnant women, with effect assessment based on outcomes observed among all the randomized fetuses after birth.

The bias illustrated in this study, which arises from restricting analyses to a very preterm population, underscores the importance of methodologic rigour in non-experimental epidemiologic research with a causal focus. The recent creation of research networks of neonatal intensive care units (NICUs) in several countries has facilitated the reporting and

benchmarking of neonatal outcomes, and research on neonatal health [45–55]. These networks have collaborated in NICU studies comparing neonatal outcomes by centre and country, and also studies of prenatal determinants [4–13, 56]. Such studies are particularly vulnerable to bias and can obscure health disparities. With study populations restricted to very preterm live births and births-based calculations of rates, NICU studies have shown lower mortality, necrotizing enterocolitis and sepsis rates among very preterm infants born to women of advanced maternal age [5]. Similarly, births-based analyses of very preterm infants have shown that hypertension in pregnancy is associated with a lower risk of death, severe brain injury and retinopathy [6]. A recent study of severe neonatal morbidity among very preterm infants also highlighted the underestimation of racial disparities in analyses that rely on births-based calculations [57]. Biased inferences from studies restricted to preterm births may be even more insidious in regional comparisons of perinatal mortality or morbidity without prior expectations regarding differences [4, 7, 58]. Although studies may be restricted to very preterm infants with the intent of comparing NICU practices (S6a Fig), the contrast of delivery hospital or regional rankings with regard to fetal and neonatal mortality at early gestational age can be similarly biased by differential distributions of gestational age at birth (S6b Fig), which is an intermediate factor between region or hospital admission and perinatal death. Distributions of gestational age at birth could differ between regions due to variations in obstetrical and neonatal practices and access to care for example. NICU comparisons can also be biased because they are limited to live-born infants who survive until admission to the NICU. Careful consideration should be given to such potential sources of bias in the interpretations of findings.

As illustrated in our study, births-based analyses restricted to very preterm births bias causal inference, as they are restricted to the left-end of the gestational age distribution. Such restriction results in a right truncation and analysis of incomplete data from the original cohort of individuals exposed or unexposed to a risk factor in pregnancy (see the S1 Appendix for an illustrated example). Births-based rates are not inherently problematic as they can be appropriate for prognostication (since they quantify the mortality experience of very preterm infants of women who are older, smoke, or have hypertension, for example). However, such rates are a poor basis for causal inference regarding prenatal factors (such as older maternal age, smoking and hypertension), addressing health inequities and evaluating neonatal health services (through between-centre and between-region comparisons). The causation vs prediction dichotomy may be best illustrated by the effect of maternal smoking on neonatal mortality [16]: it is preferable to frame the lower mortality rates among low birth weight and preterm infants of mothers who smoke in pregnancy in prognostic (predictive) terms. It is well-understood that maternal smoking has deleterious effects on the fetus and infants, irrespective of the gestational age at birth. Cohorts of very preterm births represent a select, truncated subpopulation, and studies restricted to very preterm births are unsuitable for non-experimental epidemiologic studies that attempt to estimate the causal effect of prenatal exposures using birth-based denominators.

## Strengths and limitations

Information for our study was obtained from two large data sources that have been validated for epidemiologic research purposes [30, 31]. However, some misclassification of gestational age at birth and underreporting of hypertensive disorders is likely. Hypertensive disorders in pregnancy in our study cohort may have occurred at a gestation later than 24 weeks, but in absence of information on the time of diagnosis, we were not able to treat this exposure as a time-dependent variable. However, separate analyses of chronic hypertension and other

hypertensive disorders showed the same risk of bias. We obtained a measure of the degree of confounding of the hypertension-perinatal death association by adjusting for maternal age, race and diabetes. It is possible, though not likely, that other factors such as parity, chronic diseases and socioeconomic status could have confounded the association to a greater extent since adjustment for maternal age and diabetes, two well-known confounders of the relationship between hypertensive disorders and outcomes, resulted in a 5 to 11% change in the rate ratios. Strong unmeasured confounders would be required to nullify or change the direction of the association. Information on Canadian births was obtained from hospitalization records, and early neonatal deaths that occurred after discharge could have led to a slight underestimation of the frequency of this outcome. Furthermore, data constraints led us to use information on the gestational age at birth for stillbirths, whereas gestational age at the time of fetal death would have been preferable. Although such systematic errors could have affected absolute rate estimates, they are unlikely to have occurred differentially in the contrasted populations and would not have had a major impact on our conclusions.

## Conclusion

Our analyses highlight the bias inherent in studies restricted to very preterm birth subpopulations that aim to estimate the causal effect of prenatal risk factors, social determinants of health and between-country and between-centre comparisons. Although the adverse consequences on child development and lifelong health make continued research on very preterm birth a priority, non-experimental studies on this subpopulation require to be designed and analyzed with appropriate care and attention in order to avoid erroneous inferences. A careful consideration of the study question is paramount in identifying the appropriate study population and methods of analysis.

## Supporting information

**S1 Fig.** Gestational age-specific perinatal death rates of singletons with no congenital or chromosomal anomalies by maternal race using a births-based denominator (A) or using a fetuses-at-risk denominator (B), United States, 2006–2015. The yellow area highlights the restricted subpopulation at 24–31 weeks' gestation.
(TIF)

**S2 Fig.** Gestational age-specific perinatal death rates of singletons with no congenital or chromosomal anomalies in Canada or in the United States using a births-based denominator (A) or using a fetuses-at-risk denominator (B), 2006–2015. The yellow area highlights the restricted subpopulation at 24–31 weeks' gestation.
(TIF)

**S3 Fig. Gestational age-specific birth rates of singletons with no congenital or chromosomal anomalies exposed or not to hypertensive disorders in pregnancy, United States, 2006–2015.**
(TIF)

**S4 Fig. Gestational age-specific birth rates of singletons with no congenital or chromosomal anomalies by maternal race, United States, 2006–2015.**
(TIF)

**S5 Fig. Gestational age-specific birth rates of singletons with no congenital or chromosomal anomalies in Canada and in the United States, 2006–2015.**
(TIF)

**S6 Fig.** Directed acyclic graph of relationships between NICU practices and perinatal deaths (A) and between hospital or region of residence and perinatal death (B).
(TIF)

**S1 Table. Comparisons of rates of early neonatal death rates at 24–31 weeks' gestation and overall among singletons with no congenital or chromosomal anomaly, 2006–2015.**
(PDF)

**S1 Appendix. Example of an antenatal exposure associated with gestational age at birth, resulting in a paradoxical association with mortality among very preterm births.**
(PDF)

## Author Contributions

**Conceptualization:** Amélie Boutin, K. S. Joseph.

**Formal analysis:** Amélie Boutin.

**Funding acquisition:** K. S. Joseph.

**Supervision:** K. S. Joseph.

**Writing – original draft:** Amélie Boutin.

**Writing – review & editing:** Amélie Boutin, Sarka Lisonkova, Giulia M. Muraca, Neda Razaz, Shiliang Liu, Michael S. Kramer, K. S. Joseph.

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
