## [Decision Letter · Decision Letter 0]

5 Mar 2021

PONE-D-20-40633

Bias in comparisons of mortality among very preterm births: a cohort study

PLOS ONE

Dear Dr. Boutin,

Thank you for submitting your manuscript to PLOS ONE. After careful consideration, we feel that it has merit but does not fully meet PLOS ONE’s publication criteria as it currently stands. Therefore, we invite you to submit a revised version of the manuscript that addresses the points raised during the review process.

We look forward to receiving your revised manuscript.

Kind regards,

Kelli K Ryckman

Academic Editor

PLOS ONE

Journal Requirements:

3.We note that the grant information you provided in the ‘Funding Information’ and ‘Financial Disclosure’ sections do not match.

Reviewers' comments:

Reviewer's Responses to Questions

**Comments to the Author**

1. Is the manuscript technically sound, and do the data support the conclusions?

Reviewer #1: Yes

Reviewer #2: Yes

2. Has the statistical analysis been performed appropriately and rigorously? 

Reviewer #1: Yes

Reviewer #2: Yes

3. Have the authors made all data underlying the findings in their manuscript fully available?

Reviewer #1: No

Reviewer #2: No

4. Is the manuscript presented in an intelligible fashion and written in standard English?

Reviewer #1: Yes

Reviewer #2: Yes

5. Review Comments to the Author

Reviewer #1: This is a well-written manuscript that examines the differences in how 3 risk factors (maternal hypertension, maternal race, and U.S. vs Canada country of birth) are associated with perinatal and neonatal mortality when analyzed using different subsets of births (limited to births at 24-31 weeks gestation or to all births ≥24 weeks) and with different denominators (among births at the same gestational age week or by a fetuses-at-risk approach). Although it is a methodology-focused manuscript, it is written in a language that should be accessible to both clinicians and clinical researchers. Study data are from population-based datasets based on vital statistics in the U.S. and Canada. The authors show that the methodologic choices described above (subsetting births by a gestational age range and using a denominator based on births or potential births [fetuses-at-risk]) provide paradoxical-seeming results. The findings are important, particularly, as the authors point out, because of the common use of cohort studies based on subsets of births in perinatal/neonatal medicine and epidemiology (example cohorts with hundreds of papers between them that have influenced modern obstetric/neonatology practice: Canadian Neonatal Network, NICHD Neonatal Research Network, iNEO, Vermont Oxford Network, etc). The authors should be commended for a simple demonstration of a complex subject.

MAJOR COMMENTS:

The conclusion "Studies of perinatal risk factors and between-centre or between-country comparisons of

perinatal mortality lead to biased inferences when restricted to very preterm births" seems overly simplistic. Similarly, there are several instances of this language in the Discussion (e.g., "Our study shows that comparisons of perinatal mortality between fetuses of mothers with and without hypertensive disorders of pregnancy, between mothers of different ethnicity, and between mothers in Canada and the United States, were seriously biased when the study population was restricted to very preterm births and analyses were based on births-based denominators.")

The authors might further explore the reasons for the paradoxical associations that they observe. They acknowledge the rich literature on "collider stratification bias" and the theoretical mechanistic model of the "accelerated birth rate" whereby infants in some subgroups who are at lower risk are born earlier and so these subgroup-defining traits appear protective at low gestational ages.

However, there are alternative explanations for the authors findings that do not fall squarely in the realm of bias. As a clinical neonatologist, I care for infants who are born at early gestational ages. From my point of view, among the very preterm babies admitted to our unit in any given week, birth secondary to maternal hypertensive disease is a positive prognostic sign (compared to the alternative, which is usually birth secondary to some maternal inflammatory/infectious process). Is that point of view (from the neonatologist or perinatologist who delivers or cares for the baby) not valid? The authors allude to the difference between prognostic (my perspective on how the baby will do) and causal models (concerned with etiologic risks), but this might be further explored. From the perspective of the neonatologist/perinatologist, the comparison for births born due to maternal hypertension is not healthy normal pregnancies/births - rather, the comparison is to other births due pathology.

Another alternative point of view to consider is that of effect modification. Consider the example of a persistent patent ductus arteriosus (PDA), a common presentation in premature infants that may result in morbidity. In the aggregate, persistent PDA is associated with morbidity. But, in the particular, there are conditions (e.g., transposition of the great arteries) that affect a minority of the group in which the PDA confers a major protective advantage. In a similar way, is it not possible that while, in the aggregate, birth in Canada confers a lower risk of mortality than in the U.S. -- at the same time, care for very preterm births (due to a more aggressive/intensive/expensive approach) in the U.S. (versus Canada) confers a lower risk of mortality for this minority of patients?

The authors should explore reasons for their findings that are not simply the result of bias. Bias implies that these other perspectives are erroneous, which is not clearly the case. They might consider a more nuanced elaboration of how to interpret their findings in light of the research/clinical question being asked.

MINOR COMMENTS

Ln 33 - specify that this is the perinatal (or neonatal) death rate

Ln 86 - consider removing phrase “very poor survival.” In some cohorts, survival exceeds >50% at 22 and 23 weeks. The other reasons are valid enough without this one.

Table 1 - some of the cell totals for no hypertensive disorders in pregnancy are greater than the overall totals for those rows

Table 2 - I found the inclusion of "Births- based/fetuses- at risk calculation (per 1,000 total births)" under "overall" perinatal death rate confusing. (Same applies to S1 Table.) I think that the authors are trying to demonstrate that these formulations are the same. But, because this is already specified in the text, perhaps they should simply write "Calculation per 1,000 total births" (easier to read/less confusing) and, if needed, use a footnote to remind the reader that "for the overall calculation, births-based and fetuses-at-risk formulations are equivalent."

Ln 283 - why write "though not likely"? Is that quantifiable (as in an E-value?) or is it an opinion?

Ln 293 - as noted above, perhaps "potential bias"? Or some more nuanced interpretation?

S3Fig - should the y-axis be for "fetuses-at-risk" (same as S4 and S5Fig)?

Reviewer #2: This paper is extremely well written and tackles a very important issue around bias in perinatal mortality and neonatal mortality rates. It takes the research around analysis of births versus fetuses at risk in an important direction and is written in an easy to understand style with clear diagrams.

I have only a few minor points:

1. A definition of early neonatal death and perinatal death used here would be extremely helpful as there some slight variation in the use of these terms.

2. Some of the graphs may benefit from alternative plotting shapes to add differentiation of the lines as the colours are hard to differentiate e.g. triangles and squares to differentiate between the ethnic group data that is very closely aligned

3. More detailed description of why the biases might arise would benefit the discussion and make it easier to interpret with perhaps examples.

6. PLOS authors have the option to publish the peer review history of their article (what does this mean?). If published, this will include your full peer review and any attached files.

Reviewer #1: No

Reviewer #2: **Yes: **Lucy Smith

---

## [Author Response · Author response to Decision Letter 0]

18 May 2021

The response to the reviewer has been uploaded as a word document.

---

## [Decision Letter · Decision Letter 1]

16 Jun 2021

Bias in comparisons of mortality among very preterm births: a cohort study

PONE-D-20-40633R1

Dear Dr. Boutin,

We’re pleased to inform you that your manuscript has been judged scientifically suitable for publication and will be formally accepted for publication once it meets all outstanding technical requirements.

Kind regards,

Kelli K Ryckman

Academic Editor

PLOS ONE

Additional Editor Comments (optional):

Reviewers' comments:

Reviewer's Responses to Questions

**Comments to the Author**

1. If the authors have adequately addressed your comments raised in a previous round of review and you feel that this manuscript is now acceptable for publication, you may indicate that here to bypass the “Comments to the Author” section, enter your conflict of interest statement in the “Confidential to Editor” section, and submit your "Accept" recommendation.

Reviewer #1: All comments have been addressed

2. Is the manuscript technically sound, and do the data support the conclusions?

Reviewer #1: Yes

3. Has the statistical analysis been performed appropriately and rigorously? 

Reviewer #1: Yes

4. Have the authors made all data underlying the findings in their manuscript fully available?

Reviewer #1: No

5. Is the manuscript presented in an intelligible fashion and written in standard English?

Reviewer #1: Yes

6. Review Comments to the Author

Reviewer #1: This manuscript is important and well-written. The authors quantitatively demonstrate a common and poorly understood cause of bias (at least among clinicians) pervasive in cohort studies in neonatology/perinatology. The authors should be commended on this work.

All of my comments and questions from the first review were well addressed.

A few minor things, in case they are of any value to the authors:

Ln 120 - I understand what this is intending to say, but wonder if the word “overall” is confusing. It may read to some as if there is supposed to be a comma after it. Consider alternative wording - maybe adding “The" as in "The overall…” would fix this.

Ln 245 - perhaps “effect modification, SUCH AS due …” would be more appropriate. Other arguments about mechanistic effect modification could be made.

Ln 284 - Some large cohort studies do take this into account. The authors' point is valid but doesn’t seem quite on-point and it isn't clear how this is related to the analysis.

If the authors wish to explore other common sources of biases in this literature of neonatal cohort studies limited to gestational age, they might consider also noting issues such as: non-consideration of the intention to treat (when non-treatment results in death); mixing of inborn and outborn infants without consideration of those who die prior to transfer; limitation to live births where the exposure (e.g., region/SES) may be related to classification as live birth or stillbirth. All of these (and surely others) have potentially severe results on causal inference.

I wonder if the authors would rather drop this extra comment about an unrelated source of bias to their analysis (left truncation to NICU admission) -- or, alternatively, provide a more detailed elaboration. The listing of one unrelated bias sometimes observed just seemed out of place.

Ln 328 - “require to be” = “should be”?

7. PLOS authors have the option to publish the peer review history of their article (what does this mean?). If published, this will include your full peer review and any attached files.

Reviewer #1: **Yes: **Matthew Rysavy

---

## [Editor Report · Acceptance letter]

21 Jun 2021

PONE-D-20-40633R1 

Bias in comparisons of mortality among very preterm births: a cohort study 

Dear Dr. Boutin:

I'm pleased to inform you that your manuscript has been deemed suitable for publication in PLOS ONE. Congratulations! Your manuscript is now with our production department. 

Kind regards, 

on behalf of

Dr. Kelli K Ryckman 

Academic Editor

PLOS ONE